# STRUCTURAL SEMANTIC FEATURES FOR IMPROVED AI-GENERATED FAKE IMAGE DETECTION

## ABSTRACT

The proliferation of AI-generated content (AIGC) has made the accurate detection of fake images a critical challenge. Existing state-of-the-art methods, such as PatchCraft and AIDE, primarily leverage local features like patch-wise frequency information or global semantic features derived from large-scale models like CLIP. While effective, these approaches often fail to incorporate the underlying structural semantics of an image, which are crucial for detecting the subtle inconsistencies and artifacts left by generative models. We propose a novel approach that augments existing AIGC detection frameworks by explicitly incorporating structural semantic information. Our method employs cuboidal partitioning, a hierarchical tool that recursively divides an image into meaningful sub-regions. At each division, we extract a measure of the statistical difference between the parent and child segments, which are then integrated with AIDE's existing features. Experimental results demonstrate our model's superior performance. We establish a new state-of-the-art in mean accuracy on the GenImage benchmark, proving our effectiveness on modern diffusion models. Our method also shows strong generalization by achieving second-best overall mean accuracy on the diverse AIGCDetect benchmark and a second-place finish on the challenging Chameleon dataset. These results highlight the significant value of structural semantics for building robust and generalizable AIGC detectors.

## 1 INTRODUCTION

The rapid advancement of generative models has ushered in a new era of digital media, where realistic AI-generated content (AIGC) is becoming increasingly common. Modern generative models, including both Generative Adversarial Networks (GANs) (Goodfellow et al., 2014; Zhu et al., 2017; Brock et al., 2019) and Diffusion Models (DMs) (Ho et al., 2020; Rombach et al., 2022; Song et al., 2021), have demonstrated an unprecedented ability to synthesize high-quality images that are often indistinguishable from real-world photography. While these technologies offer immense creative potential, their easy accessibility also raises serious concerns for image forensics, the fight against misinformation, and copyright protection.

Early detection methods for AIGC often relied on general-purpose deep learning backbones like ResNet-50 (He et al., 2016) and Vision Transformers (Dosovitskiy et al., 2021), which treat the detection task as a binary classification problem. While effective, these models function as black boxes, learning to identify artifacts without explicit guidance. This has led to the development of detectors that focus on **explicit feature extraction**, leveraging our domain knowledge of how generative models operate. Approaches like FreDect (Frank et al., 2020) and PatchCraft (Zhong et al., 2024) identify subtle, localized artifacts in the frequency domain or through inter-pixel correlations. Another powerful paradigm, such as the one used in the AIDE model (Yan et al., 2025), leverages a hybrid approach that combines low-level pixel statistics with high-level semantic cues from pre-trained vision models. While highly effective, these state-of-the-art detectors often overlook a crucial and complementary source of information: the **structural semantics** of an image.

Recent work by Kamali et al. (2024) provides a comprehensive taxonomy of inconsistencies in AI-generated images, categorizing them into five types, from anatomical implausibilities to violations of physics. While many existing detectors are designed to capture statistical or textural artifacts, they are less equipped to deal with these higher-level, structural inconsistencies. We argue that the

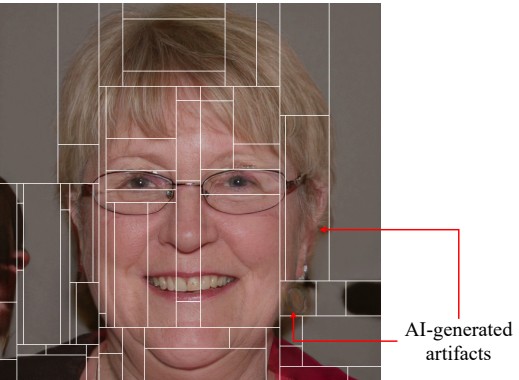

Figure 1: An example of fake image detection using the proposed structural feature-based technique, which successfully isolated AI-generated artifacts missed by the existing AIDE method.

way an image's content is organized in the scene, a fundamental principle in human vision, is often imperfectly replicated by generative models, leaving a detectable trace. This makes our method uniquely suited to address inconsistencies related to anatomical and functional implausibilities as well as violations of physics.

To illustrate the limitations of existing detectors and motivate our approach, we present a qualitative example from the challenging WFIR (West & Bergstrom, 2019) facial dataset. As shown in Fig 1, the base AIDE model, which relies on a combination of frequency and domain features, incorrectly classifies a generated face as real. In contrast, our enhanced model, incorporating our new structural features, correctly identifies the image as fake. A closer look at our method's output reveals the mechanism behind this success. The hierarchical partitioning process successfully isolated two distinct segments: one around the person's left ear and another around a hair-like structure below it. This demonstrates that our structural features can detect nuanced, localized irregularities that are often missed by human eyes and other state-of-the-art models, proving to be a critical and complementary capability for robust AI-generated content detection.

In this work, we introduce a novel approach that augments existing detection frameworks by explicitly incorporating these structural semantics. Our proposed method leverages a proven cuboidal partitioning algorithm (Ahmmed et al., 2022), previously used for building image similarity metrics (Haque et al., 2025), to recursively divide an image into statistically meaningful sub-regions. By quantifying the statistical difference at each level of this hierarchy, we generate a rich, structural feature vector that serves as a powerful new fingerprint. We integrate these features into the AIDE model, demonstrating that they are highly complementary to its existing hybrid feature set.

Our contributions are summarized as follows:

- **We present the first application of a hierarchical structural analysis method for the purpose of AIGC detection**, introducing a novel and complementary feature type to the field.
- **We establish a new state-of-the-art on the GenImage benchmark**, demonstrating that our approach is particularly effective at detecting artifacts from modern diffusion models.
- **We provide a comprehensive evaluation of our model, demonstrating its robust cross-generator and out-of-distribution generalization capabilities**. We show competitive performance on the AIGCDetect benchmark and, more importantly, prove our model's strength on the challenging, human-deceptive images of the Chameleon dataset.

## 2  RELATED WORKS

This section reviews the key literature on AI-generated image detection and structural image analysis that informs our approach.

## 2.1 AI-Generated Image Detection

The demand for effective AI-generated image detectors has grown exponentially. Early research focused on forensic cues tied to specific generative models. For instance, FreDect (Frank et al., 2020) observed artifacts in the frequency domain of GAN-generated images, attributed to the upsampling operation. CNNSpot (Wang et al., 2020) demonstrated that a classifier trained on a single GAN could generalize to others, with careful preprocessing. Other foundational approaches, such as Spec (Zhang et al., 2019) and F3Net (Qian et al., 2020), also focused on frequency and feature-level inconsistencies.

More recent approaches have explored novel perspectives to achieve superior generalization. UnivFD (Ojha et al., 2023) proposes a universal linear classifier trained on features from a pre-trained CLIP-ViT model. DIRE (Wang et al., 2023) computes a *DIRE feature* by measuring the difference between an image and its reconstruction from a pre-trained diffusion model. PatchCraft (Zhong et al., 2024) identifies a universal fingerprint across various generative models by comparing richtexture and poor-texture patches, relying on the discrepancy of inter-pixel correlations. Other notable methods include GenDet (Zhu et al., 2023a), LGrad (Tan et al., 2023), and LNP (Liu et al., 2022), which leverage various gradients, noise patterns, and neural texture cues. Approaches like Fusing (Ju et al., 2022), GramNet (Liu et al., 2020), and NPR (Tan et al., 2024) have also demonstrated strong performance by focusing on specific signal processing or neural network-based artifacts. Furthermore, general-purpose vision backbones, such as ResNet-50 (He et al., 2016), DeiT-S (Touvron et al., 2021), and Swin-T (Liu et al., 2021), have been adapted for this task, providing a baseline for feature representation quality.

The AIDE model (Yan et al., 2025), which serves as the foundation for our work, proposes a powerful hybrid approach. It combines low-level pixel statistics, derived from DCT-based (Ahmed et al., 1974) patches and Spatial Rich Model (SRM) (Fridrich & Kodovsky, 2012) filters, with high-level global semantics captured by a pre-trained CLIP (Radford et al., 2021) encoder. This mixtureof-experts approach demonstrated strong performance across multiple benchmarks. While these methods have achieved remarkable success, they tend to view images as either a collection of local patches or a global semantic unit, often missing the hierarchical structure.

## 2.2 Structural and Hierarchical Image Analysis

The analysis of an image's underlying structure and composition has a long history in computer vision. Classic methods like quad-trees and hierarchical k-means (Larose & Larose, 2014) were used to create multi-resolution representations for tasks like image retrieval and segmentation. Our work is built upon the cuboidal partitioning algorithm (Ahmmed et al., 2022), an established technique that has been used for image analysis over the years. Recently, a structural image similarity metric based on scene composition structure (Haque et al., 2025) was developed using this algorithm. The metric demonstrated that recursively partitioning an image based on statistical differences could effectively capture its intrinsic organization. In this paper, we apply this established structural analysis technique to the new domain of AIGC detection. To the best of our knowledge, we are the first to demonstrate that features derived from such a hierarchical, structural analysis can serve as a powerful fingerprint for identifying AI-generated images, particularly from modern diffusion-based models. This bridges the fields of structural image analysis and image forensics, offering a fresh perspective on a pressing problem.

## 3 Methodology

In this section, we detail our approach for enhancing AI-generated image detection by incorporating novel structural features. The proposed hybrid network architecture is depicted in Fig. 2, and its components and data flow are described in the following subsections.

### 3.1 AIDE Baseline Architecture

Our work is built upon the AIDE model, a robust hybrid detector that combines features from both low-level pixel statistics and high-level image semantics. As shown on the left of Fig. 2, the original AIDE architecture consists of two primary feature extraction modules:

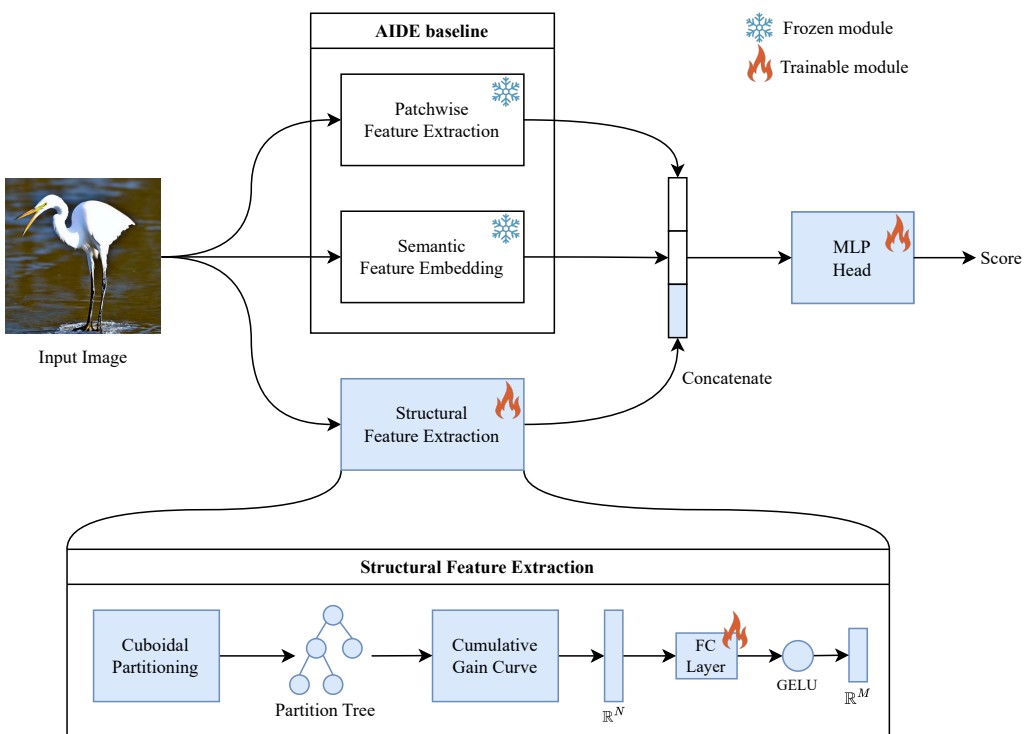

Figure 2: The proposed hybrid network architecture, which integrates a new, trainable structural feature extractor (highlighted in blue) with a pre-trained AIDE backbone.

- **Patchwise Feature Extraction:** This module captures localized statistical artifacts. It first selects the most information-rich patches from the input image using a Discrete Cosine Transform (DCT) scoring mechanism. These patches are then fed through an encoder, which includes Spatial Rich Model (SRM) (Fridrich & Kodovsky, 2012) filters, to extract a feature vector representative of low-level noise patterns and local textures.
- **Semantic Feature Embedding:** To capture high-level semantic inconsistencies, this module utilizes a pre-trained vision-language model, specifically CLIP (Radford et al., 2021). The entire image is encoded by CLIP's vision transformer, yielding a global feature vector that represents the image's overall content and context.

These two feature sets are then concatenated and passed to a final MLP head, referred to as the *Discriminator*, which is responsible for the final binary classification of the image as either real or fake.

## 3.2 STRUCTURAL FEATURE EXTRACTION VIA CUBOIDAL PARTITIONING

To create a robust detector, we propose a novel set of features that encode an image's underlying hierarchical structure. Our method leverages a recursive partitioning technique that identifies and quantifies the most prominent structural boundaries within an image.

The process begins by treating the entire image, denoted by $I$, as the initial segment. We quantify the **structural homogeneity** of this segment using the sum of squared errors (SSE) of its pixel-level features. For a given segment $S$, its SSE is defined as:

$$e_S = \sum_{p_i \in S} \|p_i - \mu_S\|^2, \tag{1}$$

where $p_i$ is the feature vector (e.g., RGB values) of the i-th pixel, and $\mu_S$ is the mean feature vector of the segment $S$.

The image is iteratively partitioned by finding the optimal axis-parallel cut, either horizontal or vertical, that maximally reduces the total SSE. This reduction, which we term the **gain** ($g$), serves as a metric for the statistical significance of a structural boundary. For a segment $S$ split into two sub-segments $S_1$ and $S_2$, the gain is calculated as:

$$g = e_S - (e_{S_1} + e_{S_2}).\qquad(2)$$

By a greedy approach, the cut that yields the highest gain, $\hat{g} = \max_{\forall \text{cuts}} g$, is selected. This process is repeated hierarchically, always selecting the sub-segment that offers the greatest potential gain for the next split. This results in a sequence of $N$ gain values, each corresponding to a progressively finer structural division of the image.

The cumulative sum of these ordered gain values forms our unique structural feature vector. This vector, with its $N$ data points, effectively encodes the image's organizational hierarchy from its coarsest to its finest details. To ensure these features are comparable across different images with varying overall structural complexity, we normalize the cumulative gain values. The i-th element of our feature vector, $\tilde{g}_i$, is given by:

$$\tilde{g}_i = \frac{1}{e_I} \sum_{j=1}^{i} \hat{g}_j, \quad 1 \le i \le N,\qquad(3)$$

where $e_I$ is the initial SSE of the full image. In our experiments, we use $N = 1024$ to capture the 1024 most significant structural boundaries.

To effectively integrate this new structural feature set with the baseline AIDE model, the resulting 1024-dimensional feature vector (Fig. 2) is first passed through a fully connected layer, followed by a GELU activation function (Hendrycks & Gimpel, 2023). This process acts as a non-linear encoder, compressing and transforming the hierarchical features into a compact $M = 256$ dimensional representation. The GELU activation is chosen for its smooth, non-monotonic properties, which are beneficial for stable learning. This compression step ensures our features are integrated efficiently, without disproportionately increasing the model's complexity.

### 3.3 FEATURE INTEGRATION AND TRAINING

As depicted in Fig. 2, our proposed features are designed to be easily integrated into the AIDE architecture in a modular fashion. The structural feature vector is simply concatenated with the existing two feature sets from the Patchwise and Semantic modules. The final, combined feature vector is then fed into the AIDE model's original Discriminator MLP head.

To ensure that the model learns to properly weigh and integrate this new information, we freeze the pre-trained weights of the Patchwise and Semantic encoders and retrain only the final Discriminator MLP from scratch alongside the structural feature extraction module. This approach allows the model to learn the optimal way to combine our novel structural features with the existing low-level and high-level features, without the need for expensive end-to-end retraining.

## 4 EXPERIMENTS

In this section, we present a comprehensive evaluation of our proposed method. We demonstrate the effectiveness of our model on several widely used public benchmarks for AI-generated image detection. Our results are compared against a range of state-of-the-art baselines to highlight the significant performance gains introduced by our structural features.

### 4.1 DETECTORS FOR COMPARISON

To provide a comprehensive evaluation, we compare our method against a wide range of state-of-the-art and foundational detectors from the existing literature. The specific baselines used vary by benchmark, as we rely on the comparison results published in the original papers. Below is a list of the key methods included in our comparison tables.

- **General Vision Backbones:** ResNet-50 (He et al., 2016), DeiT-S (Touvron et al., 2021), and Swin-T (Liu et al., 2021). These models serve as a strong baseline, demonstrating the efficacy of general image classification architectures for this specific task.

- **Forensic-based Detectors:** FreDect (Frank et al., 2020) and Spec (Zhang et al., 2019) rely on frequency domain analysis. CNNSpot (Wang et al., 2020) and F3Net (Qian et al., 2020) focus on neural artifacts and up-sampling traces.

- **Universal Detectors:** UnivFD (Ojha et al., 2023) and GenDet (Zhu et al., 2023a) are designed for broad generalization across various generative models.

- **Artifact-based Detectors:** DIRE (Wang et al., 2023) and LGrad (Tan et al., 2023) focus on reconstruction errors and gradient analysis. LNP (Liu et al., 2022) and Fusing (Ju et al., 2022) leverage noise patterns and multi-scale feature fusion.

- **Modern Approaches:** PatchCraft (Zhong et al., 2024) and NPR (Tan et al., 2024) represent recent advances, using universal artifacts and neural representations to improve detection.

- **Neural Texture-based Detectors:** GramNet (Liu et al., 2020) analyzes style-based texture features for forgery detection.

## 4.2 DATASETS AND BENCHMARKS

To thoroughly evaluate the effectiveness and generalizability of our proposed method, we conduct extensive experiments on three prominent benchmarks: GenImage (Zhu et al., 2023b), AIGCDetect (Zhong et al., 2024), and Chameleon (Yan et al., 2025). Each dataset serves a distinct purpose in assessing the capabilities of AI-generated image detectors, from measuring performance on the latest generative models to evaluating generalization on challenging, human-deceptive imagery.

**GenImage Benchmark**: GenImage is a large-scale, million-image benchmark designed to evaluate detectors on a wide range of modern, high-quality generative models. It is particularly well-suited for measuring a model's performance on the latest diffusion-based architectures. The benchmark includes images generated by eight state-of-the-art models, encompassing both established GANs and cutting-edge diffusion models. These generators are: Midjourney (Midjourney, Inc., 2022), Stable Diffusion (v1.4, v1.5) (Rombach et al., 2022), ADM (Dhariwal & Nichol, 2021), GLIDE (Nichol et al., 2022), Wukong (Gu et al., 2022a), VQDM (Gu et al., 2022b), and BigGAN (Brock et al., 2019). The inclusion of this diverse set of generators allows for a comprehensive assessment of our model's ability to identify the subtle, model-specific artifacts left by each one.

**AIGCDetect Benchmark**: AIGCDetect serves as a comprehensive benchmark for evaluating generalization across a broad spectrum of AI-generated images. Its test set features a diverse collection of images from 16 different generators, including popular GANs like ProGAN (Karras et al., 2018), StarGAN (Choi et al., 2018), StyleGAN (Karras et al., 2019), and CycleGAN (Zhu et al., 2017), as well as diffusion models. We use this benchmark to test the robustness of our model against a wide variety of artifacts and to demonstrate its ability to maintain high performance across unseen or less common generative methods. While GenImage focuses on recent models, AIGCDetect provides a valuable test of a detector's universal applicability.

**Chameleon Dataset**: The Chameleon dataset, originally introduced in the AIDE paper, represents a unique and particularly challenging benchmark for out-of-distribution generalization. Unlike other datasets where images are often generated with simple prompts and may contain obvious artifacts, the images in Chameleon are designed to be *deceptively real*. According to the dataset's creators, the AI-generated images have passed a human perception *Turing Test*, meaning human annotators frequently misclassify them as real. This makes Chameleon an ideal benchmark for evaluating a detector's ability to identify subtle forgery traces that would be missed by human inspection. Our evaluation on this benchmark provides a realistic measure of how well our model performs in scenarios where detectors are most needed.

## 4.3 TRAINING AND EVALUATION

For the evaluation of our model, we follow the established methodologies for each benchmark dataset to ensure a fair and direct comparison with state-of-the-art methods. The training and evaluation procedures for each dataset are detailed below.

Table 1: Performance comparison of the proposed method against some existing methods on the GenImage (Zhu et al., 2023b) benchmark datasets, where the best and second-best results in each column are marked in **bold** and underline, respectively.

| Method | Midjourney | SD v1.4 | SD v1.5 | ADM | GLIDE | Wukong | VQDM | BigGAN | Mean |
|---|---|---|---|---|---|---|---|---|---|
| ResNet-50 | 54.90 | **99.90** | 99.70 | 53.50 | 61.90 | 98.20 | 56.60 | 52.00 | 72.09 |
| DeiT-S | 55.60 | **99.90** | 99.80 | 49.80 | 58.10 | 98.90 | 56.90 | 53.50 | 71.56 |
| Swin-T | 62.10 | **99.90** | 99.80 | 49.80 | 67.60 | 99.10 | 62.30 | 57.60 | 74.78 |
| CNNSpot | 52.80 | 96.30 | 95.90 | 50.10 | 39.80 | 78.60 | 53.40 | 46.80 | 64.21 |
| Spec | 52.0 0 | 99.40 | 99.20 | 49.70 | 49.80 | 94.80 | 55.60 | 49.80 | 68.79 |
| F3Net | 50.10 | **99.90** | **99.90** | 49.90 | 50.00 | **99.90** | 49.90 | 49.90 | 68.69 |
| GramNet | 54.20 | 99.20 | 99.10 | 50.30 | 54.60 | 98.90 | 50.80 | 51.70 | 69.85 |
| DIRE | 60.20 | **99.90** | 99.80 | 50.90 | 55.00 | 99.20 | 50.10 | 50.20 | 70.66 |
| UnivFD | 73.20 | 84.20 | 84.00 | 55.20 | 76.90 | 75.60 | 56.90 | **80.30** | 73.29 |
| GenDet | **89.60** | 96.10 | 96.10 | 58.00 | 78.40 | 92.80 | 66.50 | 75.00 | 81.56 |
| PatchCraft | 79.00 | 89.50 | 89.30 | 77.30 | 78.40 | 89.30 | 83.70 | 72.40 | 82.36 |
| AIDE | 79.38 | 99.74 | 99.76 | 78.54 | 91.82 | 98.65 | 80.26 | 66.89 | 86.88 |
| Ours | 82.04 | 99.83 | 99.75 | **81.53** | **95.18** | 99.40 | **85.09** | 73.64 | **89.56** |

**GenImage Benchmark**: To evaluate performance on GenImage, our model was trained on the Stable Diffusion v1.4 training dataset. This approach aligns with the standard procedure outlined in the original GenImage paper and is a common practice in subsequent works. Training was performed using a learning rate of 1e-5 and a batch size of 32 over 5 epochs on a single A100 GPU, which took around 15 hours to complete. The trained model was then evaluated on the test sets from all generators included in the GenImage benchmark.

**AIGCDetect Benchmark**: Our evaluation on AIGCDetect follows a similar standard methodology. The model was trained on the ProGAN training dataset. For this benchmark, training was conducted with a learning rate of 1e-5 and a batch size of 32 for a single epoch on a single A100 GPU, which took around 3 hours to complete. The final evaluation was performed across all generators within the AIGCDetect benchmark to provide a comprehensive assessment of the model's capabilities.

**Chameleon Dataset**: For the Chameleon dataset, we utilized the two pre-trained models from the previous evaluations. The first model was trained on the Stable Diffusion v1.4 dataset (as used for GenImage), and the second model was trained on the ProGAN dataset (as used for AIGCDetect). These two models were used to perform the evaluation on the Chameleon dataset, allowing us to assess their performance on this unique benchmark without additional training.

## 4.4 MAIN RESULTS ON THE GENIMAGE BENCHMARK

Our primary results on the GenImage benchmark are summarized in Table 1. This benchmark is particularly valuable for evaluating performance on modern, high-quality diffusion-based models, and it is here that our model demonstrates its most significant strength. As shown in the table, our method achieves a new state-of-the-art (SOTA) mean accuracy of **89.56%**, surpassing the previous AIDE baseline by a substantial margin of 2.68%. A closer look at the per-generator performance reveals that our approach is consistently superior with a maximum improvement of 6.75% on the BigGAN dataset, where AIDE is very weak. Our model achieves the highest accuracy on four distinct generators: ADM, GLIDE, VQDM, and Wukong. This is particularly noteworthy as these are among the most recent and powerful diffusion models, which the artifacts in our structural features are highly adept at identifying. Furthermore, our model secures the second-best performance on a number of other critical sub-benchmarks, including Midjourney and SD v1.4. This consistent performance across a diverse range of generators, where our model is either the best or the second-best in most categories, underscores the robustness and generalizability of our proposed structural features.

## 4.5 MAIN RESULTS ON THE AIGCDETECT BENCHMARK

While the AIGCDetect benchmark includes a broader range of older GAN-based models, our method remains highly competitive, as detailed in Table 2. Our model achieves a mean accuracy of 91.85%, which is the second-best overall and only slightly behind the AIDE baseline, reinforc-

ing the effectiveness of our features in a broader context. A more granular analysis of the results confirms that our structural features are a crucial and complementary addition. Our model achieves state-of-the-art performance on several key subsets, including StarGAN, StyleGAN, and WFIR. This is particularly significant as our SOTA result on WFIR, a dataset composed of human faces, reinforces our finding from Fig. 1, that our structural features are highly effective at detecting subtle AI-generated artifacts in human faces. This also suggests that our features are especially effective at detecting the structural discrepancies in high-quality architectures like StyleGAN and its variants. Additionally, we are the second-best performer on VQDM, Wukong, and DALLE2, reinforcing the fact that our model's performance on a wide range of generators is not a mere coincidence.

Table 2: Performance comparison of the proposed method against some existing methods on the AIGCDetect benchmark (Zhong et al., 2024) datasets, where the best and second-best results in each column are marked in **bold** and underline, respectively.

| Method | ProGAN | StyleGAN | BigGAN | CycleGAN | StarGAN | GauGAN | StyleGAN2 | WFIR | ADM | Glide | Midjourney | SD v1.4 | SD v1.5 | VQDM | Wukong | DALLE2 | SDXL | Mean |
|---|---|---|---|---|---|---|---|---|---|---|---|---|---|---|---|---|---|---|
| CNNSpot | **100.00** | 90.17 | 71.17 | 87.62 | 94.60 | 81.42 | 86.91 | 91.65 | 60.39 | 58.07 | 51.39 | 50.57 | 50.53 | 56.46 | 51.03 | 50.45 | 53.03 | 69.73 |
| FreDect | 99.36 | 78.02 | 81.97 | 78.77 | 94.62 | 80.57 | 66.19 | 50.75 | 63.42 | 54.13 | 45.87 | 38.79 | 39.21 | 77.80 | 40.30 | 34.70 | 51.23 | 63.28 |
| Fusing | **100.00** | 85.20 | 77.40 | 87.00 | 97.00 | 66.80 | 83.30 | 66.80 | 49.00 | 57.20 | 52.20 | 51.00 | 51.40 | 55.10 | 51.70 | 52.80 | 55.60 | 67.63 |
| LNP | 99.67 | 91.75 | 77.75 | 84.10 | 99.92 | 75.39 | 94.64 | 70.85 | 84.73 | 80.52 | 65.55 | 85.55 | 85.67 | 74.46 | 82.06 | 88.75 | 87.75 | 84.07 |
| LGrad | 99.83 | 91.08 | 85.62 | 86.94 | 99.27 | 78.46 | 85.32 | 55.70 | 67.15 | 66.11 | 65.35 | 63.02 | 63.67 | 72.99 | 59.55 | 65.45 | 71.30 | 75.11 |
| UnivFD | 99.81 | 84.93 | 95.08 | 98.33 | 95.75 | **99.47** | 74.96 | 86.90 | 66.87 | 62.46 | 56.13 | 63.66 | 63.49 | 85.31 | 70.93 | 50.75 | 50.73 | 76.80 |
| DIRE-G | 95.19 | 83.03 | 70.12 | 74.19 | 95.47 | 67.79 | 75.31 | 58.05 | 75.78 | 71.75 | 58.01 | 49.74 | 49.83 | 53.68 | 54.46 | 66.48 | 55.35 | 67.90 |
| DIRE-D | 52.75 | 51.31 | 49.70 | 49.58 | 46.72 | 51.23 | 51.72 | 53.30 | 98.25 | **92.42** | 89.45 | 91.24 | 91.63 | 91.90 | 90.90 | 92.45 | 91.28 | 72.70 |
| PatchCraft | **100.00** | 92.77 | **95.80** | 70.17 | 99.97 | 71.58 | 89.55 | 85.80 | 82.17 | 83.79 | **90.12** | **95.38** | **95.30** | 88.91 | 91.07 | **96.60** | **98.43** | 89.85 |
| NPR | 99.79 | 97.70 | 84.35 | 96.10 | 99.35 | 82.50 | 98.38 | 65.80 | 69.69 | 78.36 | 77.85 | 78.63 | 78.89 | 78.13 | 76.11 | 64.90 | 94.10 | 83.57 |
| AIDE | 99.99 | 99.64 | 83.95 | **98.48** | 99.91 | 73.25 | 98.00 | 94.20 | 93.43 | **95.09** | 77.20 | 93.00 | 92.85 | **95.16** | 93.55 | **96.60** | **97.05** | **93.02** |
| Ours | 99.99 | **99.74** | 79.98 | 96.75 | **100.00** | 69.81 | **98.53** | **96.80** | 92.99 | 93.03 | 75.92 | 90.83 | 90.63 | 94.03 | 91.77 | 95.00 | 95.58 | 91.85 |

## 4.6 GENERALIZATION ON THE CHAMELEON BENCHMARK

The Chameleon benchmark is designed to test a detector's ability to generalize to out-of-distribution, human-deceptive images. As shown in Table 3, while our model is not the outright best performer, it consistently achieves the second-best results in both the ProGAN (58.91%) and SD v1.4 (61.39%) training scenarios. This is a crucial validation of our approach, as it demonstrates that the structural features learned by our model are not overfitting to the training distribution. Instead, they provide robust, generalizable cues that enable our detector to maintain competitive performance on a particularly challenging evaluation set, a key requirement for real-world applications.

Table 3: Performance comparison of the proposed method against some existing methods on the Chameleon (Yan et al., 2025) dataset, where the best and second-best results in each column are marked in **bold** and underline, respectively.

| Method | Training Dataset | |
|---|---|---|
| | **ProGAN** | **SD v1.4** |
| **CNNSpot** | 56.94 | 60.11 |
| **FreDect** | 55.62 | 56.86 |
| **Fusing** | 56.98 | 57.07 |
| **GramNet** | **58.94** | 60.95 |
| **LNP** | 57.11 | 55.63 |
| **UnivFD** | 57.22 | 55.62 |
| **DIRE** | 58.19 | 59.71 |
| **PatchCraft** | 53.76 | 56.32 |
| **NPR** | 57.29 | 58.13 |
| **AIDE** | 58.37 | **62.60** |
| **Ours** | 58.91 | 61.39 |

## 4.7 QUALITATIVE RESULTS

To complement our quantitative findings, we provide a qualitative analysis of our model's performance in Fig. 3, showcasing its ability to correctly classify images that the AIDE baseline misiden-

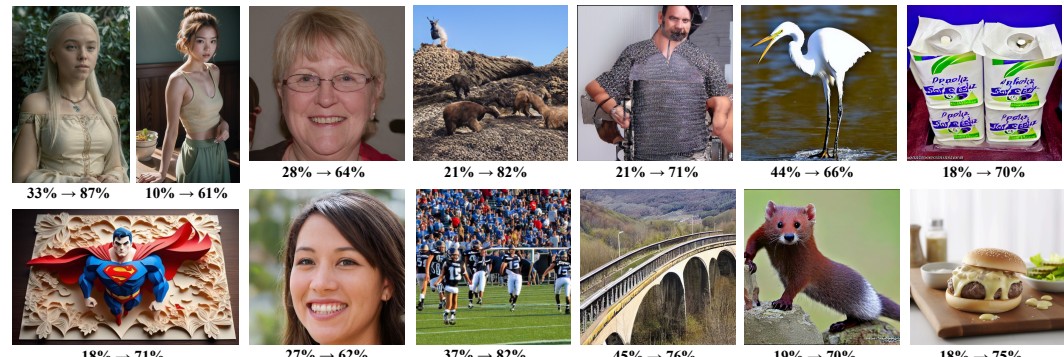

Figure 3: A comparison of model predictions on fake images where the AIDE baseline failed; scores below each image show the shift from AIDE's low confidence (<50%) to our model's high confidence (>50%).

tifies. The figure presents 13 examples of AI-generated images from the GenImage, AIGCDetect, and Chameleon datasets, where AIDE's confidence score was below 50% (indicating a *real* classification), while our model achieved a confidence of over 50% (correctly identifying the image as *fake*). The scores below each image visually demonstrate this critical shift in prediction confidence.

These examples represent the challenging cases and *blind spots* of the AIDE baseline. While the images are of high quality, they contain subtle inconsistencies that are not easily detected by features based on simple frequency artifacts or global semantics. Our model's consistent success in these cases provides compelling evidence that the proposed structural features are sensitive to these nuances, capturing flaws in composition, object integrity, or visual plausibility that a flat-feature approach overlooks. The figure thus serves as a powerful visual argument for the value of our structural approach as a crucial and complementary addition to existing detection frameworks.

### 4.8 SUMMARY OF EXPERIMENTAL FINDINGS

The experimental results on all three benchmarks consistently demonstrate the value of our proposed structural features. Our method sets a new SOTA on the GenImage benchmark, proving its superiority on modern diffusion models. On the broader AIGCDetect benchmark, our model is highly competitive overall and achieves SOTA results on critical subsets. Our strong second-best performance on the challenging Chameleon benchmark further validates the generalizability of our approach. These results collectively show that by extending the AIDE baseline with our hierarchical structural features, we have created a more powerful and robust detector.

However, a more nuanced analysis reveals that augmenting a powerful hybrid model does not guarantee universal improvement. Consistent with established findings on complex ensemble models (mixture-of-experts), which show that adding a new expert can sometimes lead to performance degradation if its contribution is not sufficiently valuable (Hansen & Salamon, 1990), we observed that our model's performance slightly decreased on certain subsets. We hypothesize that these datasets contain fewer of the structural inconsistencies or artifacts that our expert is designed to detect. In such cases, the output of our structural extractor may act as noise to the final classifier, reducing the overall accuracy. This suggests that the value of our structural features is highly context-dependent, providing significant gains in scenarios where other features fail to capture key anomalies.

## 5 CONCLUSION

This work provides strong evidence that hierarchical structural analysis offers a powerful new perspective on AI-generated content detection. By successfully augmenting an existing hybrid framework, we have shown that such structural features are highly complementary to existing approaches. A key direction for future work is to develop more adaptive feature ensemble techniques that can dynamically weigh the contribution of each expert.

## REPRODUCIBILITY STATEMENT

To ensure the reproducibility of our work, our code and model weights will be made publicly available upon acceptance. All experimental details, including hyperparameters and training time, are provided in Section 4.2 and 4.3.

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
