# OpenReview forum: "Structural Semantic Features for Improved AI-Generated Fake Image Detection"
_ICLR.cc/2026/Conference — ICLR 2026 Conference Withdrawn Submission_

### Official Review · Reviewer_j2af · 2025-10-30

**Soundness:** 1
**Presentation:** 2
**Contribution:** 1
**Rating:** 2
**Confidence:** 5

**Summary:**

The authors tackle fake image detection by building upon a popular baseline AIDE. This baseline uses a mixture of low-level and high-level features of an image to classify it as real or fake. The authors propose using an additional set of "structural" features of an image. To compute these features an image is recursively partitioned into smaller segments so that the "structural homogeneity" increases in ensuing segments. At each step, the increase in the homogeneity is measured. The final feature used is the collection of such increases at each step. Adding these features to those used by AIDE method, the authors train their fake detector which sometimes leads to an increase in performance.

**Strengths:**

- The paper uses an idea that could potentially be interesting - if one wants to break down an image recursively into its constituent segments using some appropriate function, the way a real image will be broken down versus the way an AI image will be broken down will be different. For example, maybe the AI image's segments have a high variance in their size (a combination of big and small-sized segments) whereas the real image has somewhat similar sized segments. However, the way this idea is implemented in this work has problems; please see the weaknesses.

- The paper is well written and easy to understand.

- The authors have reported results on multiple benchmarks.

**Weaknesses:**

- I did not find any satisfactory explanation on why the structural features, as defined in Section 3.2, should be particularly useful for fake image detection. The implicit assumption by the authors is that the way a real image will be partitioned using the greedy approach (line 223) will be different to the way an AI generated image will be partitioned. But even if that turned out to be true as an emergent behavior of fake images, there is nothing in the paper which hints at that. An example demonstration would be to show how the nature of partitions (e.g., the partition sizes) is different between real and fake images. But there is no qualitative or quantitative result indicating that.

- Furthermore, there is a key question about Figure 1. The approach described in Figure 2 only gives a score (e.g., between 0 and 1) which tells us whether the image is fake. I do not see how you can trace that score back to figure out which specific segments of the input image were fake. Specifically, in line 79-80, the authors mention that “... our method’s output reveals the mechanism behind this success”. How do we know that it was those specific patches (marked by red line in Fig. 1) which led to the image being classified as fake? In fact, I can find a few others which could be deemed having some artifact.

- In the results section, there are many instances of unsubstantiated claims. I discuss a few examples below
    - Line 381-383: It is not clear how the proposed method can help in localizing which of the N segments have artifacts. Is that done post-hoc by a human or does the algorithm itself indicate which of the N segments is more fake?
    - Line  376-378: I do not think the results of Table 2 “reinforce the effectiveness of our features in a broader context” because the proposed approach, being built on top of the AIDE model, achieves slightly worse performance than AIDE. Therefore, it is not clear in what way the newly added features are actually useful.
    - Line 407-409: “This is a crucial validation of our approach, as it demonstrates that the structural features learned by our model are not overfitting to the training distribution” - No. The results do not demonstrate that. Same reason as above.

- Section 4.7: It is not clear what the main purpose of this section is. As I understand, the authors were able to find some images in the test sets where the AIDE model did not give the correct label (score for fake < 50%) but their model did give the accurate label (score >50%). But given the results in Table 1, 2 and 3, it seems as if, overall, the performance of the proposed method, which is built on top of AIDE, is pretty much the same as AIDE. So, it seems natural to suppose that there are also images where the AIDE gives accurate predictions and the proposed method does not. And if so, one can create another figure like Fig. 3 showing the opposite. Are the authors claiming that there are not fake images where their model produces a low score (<50%) and the AIDE produces a high score (>50%)?

**Questions:**

Line 223: Would finding all the cuts and taking the maximum from them be very expensive? How is that done in practice? For a segment of H x W, do you iterate over all the H + W possible cuts?

Line 81: “left” should be “right”

---

### Official Review · Reviewer_SBQS · 2025-10-31

**Soundness:** 1
**Presentation:** 2
**Contribution:** 1
**Rating:** 2
**Confidence:** 5

**Summary:**

This work presents the hypothesis that existing methods do not leverage hierarchical structural information and therefore have problems with generalization. The method employs cuboidal partitioning, which gathers hierarchical information by dividing the image into meaningful subregions and uses these features alongside features from an existing method (AIDE) to improve detector performance on popular existing benchmarks.

**Strengths:**

1. Achieves SOTA/close to SOTA on popular benchmarks outperforming several popular baselines.

**Weaknesses:**

1. There is not enough evidence presented to argue that neural networks do not track hierarchical information. CNNs are in fact designed in a manner which tracks hierarchical information. This claim needs further verification.
2. For me the main concern is the fact that this method performs worse than AIDE (the baseline that it augments with structural semantic features) in this cases. This raises doubts on whether the newly incorporated features actually help. More specifically, in Tab 2,3 AIDE outperforms the proposed method on DALLE-2, SDXL, Wukong etc. This could imply that the structural sematic features provide spurious signals for detecting images from more recent, higher quality diffusion models.
3. The sensitivity of the detector to common post-processing operations (JPG, upsizing, downsizing, blur, WEBP etc) should be studied with thorough sweeps. It is important to verify this against other baselines. This would help verify if the detector is suited for detecting fake images in the wild.
4. It would also be insightful if the authors would report the AP values, in addition with accuracies, so that it would provide insights regarding the full potentials of the various detectors.
5. Line 271: I am not convinced by the classification of such general-purpose feature extractors (except for CLIP) as strong baselines. This is mainly due to their demonstrated performance, as well as the fact that people typically do not use these models directly for fake image detection without fine-tuning (How are these methods trained for fake image detection?)
6. Line 380: The paper mentions that StarGAN and other datasets on which the proposed method outperforms the other method are "key" subsets. However, it is not clear what makes these subsets more important. Further clarification on this would be helpful.

**Questions:**

Repeat of Concerns raised above,
1. What is the evidence for the claim that conventional networks do not track hierarchical information?
2. How are the general purpose feature extractors (refer weakness number 5) trained.
3. Why are the StarGAN, StyleGAN and WFIR classified as key subsets?

---

### Official Review · Reviewer_HLnj · 2025-10-31

**Soundness:** 2
**Presentation:** 2
**Contribution:** 1
**Rating:** 2
**Confidence:** 5

**Summary:**

This paper proposes a hierarchical structural analysis for AIGC detection, which recursively divides an image into statistically sub-regions. By quantifying the statistical difference at each level, a rich, structural feature vector is generated which serves as a powerful new fingerprint. Then the powerful fingerprints are integrated into the AIDE model, leading to improved performance. Experiments on GenImage and AIGCDetect benchmarks demonstrate the effectiveness of proposed method.

**Strengths:**

1. This paper proposes to address the fake image detection task from a hierarchical semantic patch level, which makes sense and is technically sound.
2. Experiments on public datasets demonstrate the effectiveness of the proposed method.
3. The authors provide case study in Fig. 3 compared with AIDE baseline, which benefits for understanding the results.

**Weaknesses:**

1. Although I can understand the authors' motivation from their descriptions in the Introduction part, the Fig. 1 actually confuses me. I suggest the authors elaborate the motivation with Fig.1.
2. The authors incorporate three different pathways for final detection. I suggest making Fig. 2 clearer since it is hard to understand what the pathway means. And there are no ablations on each part to justify which one is actually working. More analysis is needed.
3. Can the authors provide some examples on how the partition tree working?
4. The AIDE is the main baseline this paper mentions, which make this paper more like an incremental work based on AIDE. More discussions on the difference are needed.
5. The authors should compare more recent works in Tab.1, such as those published in 2024 and 2025.

**Questions:**

Please refer to the weakness part. I have many concerns about the method, experiments, and novelty. So I currently lean towards negative ratings. If the authors could address my concerns properly, I will change my ratings.

---

### Official Review · Reviewer_W9Qh · 2025-11-04

**Soundness:** 3
**Presentation:** 3
**Contribution:** 2
**Rating:** 4
**Confidence:** 4

**Summary:**

The paper proposes a new feature extraction mechanism for detecting AI-generated (fake) images, focusing on structural semantics — the hierarchical organization of image content. The method extends the AIDE model by introducing a cuboidal partitioning algorithm, which recursively divides images into sub-regions and computes a statistical gain curve representing structural homogeneity differences.
Empirical results across GenImage, AIGCDetect, and Chameleon benchmarks show good performance of proposed method.

**Strengths:**

- The proposed method introduces hierarchical structural semantics, which is well-motivated and relatively novel.
- The recursive cuboidal partitioning and statistical gain formulation are mathematically clear. The algorithm is interpretable and modularly integrates with existing backbones.

**Weaknesses:**

- While the feature itself is novel, the integration (concatenation + MLP) is straightforward. The work could benefit from exploring more adaptive fusion mechanisms.
- There is no ablation isolating the structural feature module’s contribution beyond the combined system. Reporting results with and without structural features would better quantify their standalone value.

**Questions:**

Why was simple concatenation chosen over adaptive fusion (e.g., attention weighting or learned gating) for integrating structural features? Could the latter mitigate the “context-dependent” degradation you mentioned?
How does the cuboidal partitioning scale with image resolution? What is the inference time overhead compared to baseline AIDE?

---

### Note · Authors · 2025-11-14

**Comment:**

Thank you very much for your time and the highly constructive feedback provided on our submission.

We appreciate the thorough assessments from all reviewers, who highlighted critical areas for improvement. To address these significant and insightful suggestions properly, we have decided to withdraw our submission from the current review cycle.

We are committed to incorporating your feedback to develop a much stronger and more comprehensive version of this paper for future submission.

Thank you once again for your time and efforts.

**Withdrawal Confirmation:**

I have read and agree with the venue's withdrawal policy on behalf of myself and my co-authors.